# Application of elastic net regression for modeling COVID-19 sociodemographic risk factors

Tristan A. Moxley[1,2]*, Jennifer Johnson-Leung[2,3], Erich Seamon[3], Christopher Williams[2], Benjamin J. Ridenhour[1,2,3]

**1** Bioinformatics and Computational Biology Program, University of Idaho, Moscow, ID, United States of America, **2** Department of Mathematics and Statistical Science, University of Idaho, Moscow, ID, United States of America, **3** Institute for Modeling Collaboration and Innovation, Moscow, ID, United States of America

* tmoxley@uidaho.edu

## Abstract

### Objectives

COVID-19 has been at the forefront of global concern since its emergence in December of 2019. Determining the social factors that drive case incidence is paramount to mitigating disease spread. We gathered data from the Social Vulnerability Index (SVI) along with Democratic voting percentage to attempt to understand which county-level sociodemographic metrics had a significant correlation with case rate for COVID-19.

### Methods

We used elastic net regression due to issues with variable collinearity and model overfitting. Our modelling framework included using the ten Health and Human Services regions as submodels for the two time periods 22 March 2020 to 15 June 2021 (prior to the Delta time period) and 15 June 2021 to 1 November 2021 (the Delta time period).

### Results

Statistically, elastic net improved prediction when compared to multiple regression, as almost every HHS model consistently had a lower root mean square error (RMSE) and satisfactory $R^2$ coefficients. These analyses show that the percentage of minorities, disabled individuals, individuals living in group quarters, and individuals who voted Democratic correlated significantly with COVID-19 attack rate as determined by Variable Importance Plots (VIPs).

### Conclusions

The percentage of minorities per county correlated positively with cases in the earlier time period and negatively in the later time period, which complements previous research. In contrast, higher percentages of disabled individuals per county correlated negatively in the earlier time period. Counties with an above average percentage of group quarters experienced

**Data Availability Statement:** The data that support the findings of this study are available in Figshare at 10.6084/m9.figshare.21777674. These data were derived from the following resources available

in the public domain: CDC/ATSDR (https://www.atsdr.cdc.gov/placeandhealth/svi/data_documentation_download.html), New York Times (https://github.com/nytimes/covid-19-data), and CNN (https://www.cnn.com/election/2020/results/president).

**Funding:** TAM, JJL, ES, and BJR received funds via NIH (National Institutes of Health; http://www.nih.gov) grant number 3P20GM104420-06A1S1. The funders had no role in data collection, analysis, decision to publish, or preparation of the manuscript.

**Competing interests:** The authors have declared that no further competing interests exist.

a high attack rate early which then diminished in significance after the primary vaccine roll-out. Higher Democratic voting consistently correlated negatively with cases, coinciding with previous findings regarding a partisan divide in COVID-19 cases at the county level. Our findings can assist regional policymakers in distributing resources to more vulnerable counties in future pandemics based on SVI.

## Introduction

Severe Acute Respiratory Syndrome Coronavirus 2 (SARS-CoV-2) has impacted the world since its emergence in December of 2019, with a global case total of approximately 631 million cases and global death total of approximately 6.6 million deaths as of November 2022 [1]. Effective measures to quell the pandemic and minimize hospitalizations and deaths have included quarantining, mask mandates, social distancing measures [2], and vaccines [3]. The COVID-19 pandemic is unusual in the typical epidemiological sense, as wealthier countries with better health infrastructure have higher reported attack rates than their less wealthy counterparts [4]. The heightened attack rate in the United States is a cause for concern. The U.S. has the highest confirmed case count of any nation at 97.6 million cases (as of November 2022), more than doubling India (the second-highest total) [1]. Disparities in case reporting may influence this trend. However, it is still reasonable to expect that nations with more wealth and better access to prophylactic resources would have lower disease burden overall, given the effectiveness of vaccines and other preventative tools [5].

From the perspective of the host-agent-environment model [6], social factors present as a natural explanation for some of these discrepancies. When considering social factors, it is important to make the distinction between intrinsic factors (i.e., factors that one cannot easily change; these may include race, ethnicity, socioeconomic status, etc.), and extrinsic factors (i.e., qualities and behaviors that one acquires or changes throughout their daily lives, such as political ideology, occupation, etc.). In cases where intrinsic factors correlate with cases, state or federal governments can allocate more resources to particular regions where more disadvantaged individuals may live based on pre-existing county-level risk evaluation metrics such as the Social Vulnerability Index (SVI) [3]. In cases where extrinsic factors correlate with cases, state or federal governments can assist in improving infrastructure and protocols for at-risk demographics within more vulnerable regions, as well as informing the public of proper pandemic responses through reputable sources such as the Centers for Disease Control (CDC).

Intrinsic social factors are multi-factorial in their impact on COVID-19 spread [7]. Studies have shown an adverse relationship between being a racial or ethnic minority (e.g., African American, Indigenous, Hispanic, etc.) and contracting COVID-19; this translates as higher incidences of severe cases and deaths in certain demographics [8]. The increased disease burden on racial/ethnic minorities has several contributing factors. Some of the underlying trends stem from higher rates of comorbidity, living in more crowded living conditions [7], and decreased ability to social distance due to working lower-paying, "essential" jobs in retail, transportation, agriculture, etc. [8]. Many of these disparities have been well-documented prior to COVID-19. African Americans are eight times more likely to contract HIV compared to Caucasians on average, yet coverage of pre-exposure prophylaxis for treating HIV is seven times higher in Caucasians than in African Americans [9]. Given the history of healthcare disparity and comorbidity influencing epidemiological attack rate [7], it is important to identify and prioritize these groups for prophylactic resource allocation.

Political affiliation has been hypothesized to have affected the spread of COVID-19 in the U.S., as pandemic response rapidly became a highly-politicized in the spring of 2020. This politicization was accompanied by the rapid spread and endorsement of misinformation regarding the severity and origin of SARS-CoV-2 [10]. At the state level, jurisdictions with Democrat leadership had more aggressive responses to COVID-19 on average [11]. Some states with Republican administrations—such as Idaho—were more socially lax with mask and vaccine mandates, and subsequently suffered high spikes in cases and deaths from Delta and Omicron [12]. Some ideological conservatives and religious fundamentalists "may see [scientific] experts as threatening to their social identities" [13], and thus may respond to COVID-19 in a more lax and risk-prone manner. Furthermore, Trump's higher-than-average approval among conservative citizens may have had additional effects [14]. Conservative individuals who cited former president Donald Trump and his task force as their primary information outlet were far less likely to get vaccinated [15]. However, as vaccine hesitancy has historical connections to both extremes of the political spectrum, a more formal analysis is necessary to determine the extent to which political affiliation is a contributing factor towards COVID-19 spread or vaccine acceptance [16].

Our research looks into the effect of the combination of social forces, both intrinsic and extrinsic, and demography on COVID-19 cases. The remainder of this paper is organized as follows: A brief mention of model development is provided, along with elaboration on data selection. Significant variables found through these analyses are shown and interpreted through the lens of pandemic response at the county level. Our results can inform pandemic resource allocation based on areas at higher social risk.

## Materials and methods

We analyze the relationship of several measures of social vulnerability along with 2020 presidential voting preference on the incidence of COVID-19 at the county level. Data on social measures were obtained from the CDC's Social Vulnerability Index (SVI) [17]. SVI is a percentage-per-county metric which synthesizes 15 census variables into one vulnerability score. This index is typically used to allocate necessary resources to vulnerable counties during disaster responses. Democratic voting percentage from the 2020 presidential election [18] is used as a proxy for political ideology. The full set of explanatory data is given in Table 1. Our response variable is cumulative cases per 1000 individuals and was sourced from the New York Times [1].

Elastic net regression is chosen for this analysis [19]. Our exploratory analysis using multiple regression revealed both multicollinearity and model overfitting, likely due to social factor comorbidity and varying national-level testing efforts [7, 20]. Elastic net regression is capable of overcoming both issues via regularization. We opted against using Variance Inflation Factor (VIF) to correct collinearity due to concerns of removing variables of interest from the analyses [21].

All analyses were performed using R version 4.1.3 [22]. The underlying model equation is $\mathbf{Y} = \mathbf{X}\beta + \epsilon$. In the case of elastic net regression, the estimated regression coefficients $\beta$ are determined by minimizing a penalized sum of squares. Specifically, the sum of squares is penalized by the *elastic net penalty* comprised of the $l_1$ and squared $l_2$ norm, and is dependent on hyperparameters $\alpha$ and $\lambda$. Here, $\alpha$ controls the $l_1/l_2$ mixing percentage, and $\lambda$ alters the penalization weight. A more detailed explanation of the elastic net model is available in S1 Appendix. The **caret** package was employed for determining the optimal hyperparameters necessary for constructing the elastic net models (S2 Fig displays an example of this penalization process). Verification of these parameters is determined via 70/30 repeated cross-validation with 15 iterations.

**Table 1. Explanatory variable abbreviations and their descriptions for SVI and voting percentage.**

| Variable | Description |
|---|---|
| POV | Percentage of individuals below poverty estimate |
| UNEMP | Percentage of unemployed individuals |
| NOHSDP | Percentage of individuals with no high school diploma |
| AGE65 | Percentage of individuals over the age of 65 |
| AGE17 | Percentage of individuals under the age of 17 |
| DISABL | Percentage of individuals with a non-institutionalized disability |
| SNGPNT | Percentage of individuals who are single parents with a child below the age of 18 |
| MINRTY | Percentage of minorities (all persons except white, non-Hispanic) |
| LIMENG | Percentage of individuals over the age of 5 who speak English "less than well" |
| MUNIT | Percentage of housing in structures with 10 or more units |
| MOBILE | Percentage of mobile homes |
| CROWD | Percentage of occupied housing units with more people than rooms |
| NOVEH | Percentage of households with no vehicle available |
| GROUPQ | Percentage of persons in group quarters |
| pct | Percentage voting Democratic in 2020 Presidential Election |

We model COVID-19 case rates at the county level for the ten Department of Health and Human Services (HHS) regions and two time periods. The definition of the HHS regions is given in Table 2 [23]. We utilize HHS regions for submodeling as a proxy to healthcare infrastructure, as it is known to vary across the U.S. and will subsequently provide nuance into region-based social risk factors [24]. The first time period analyzed is 22 March 2020–15 June 2021, which we will refer to as "pre-Delta" hereafter. The second time period is 16 June 2021–1 November 2021, which we will refer to "Delta" hereafter. Thus we have 20 different models of case rates in the U.S. (10 regions × 2 time periods). For each model, we calculate the root mean squared error (RMSE) for both multiple regression (as a baseline) and elastic net regression to verify that elastic net is producing more robust models. In addition to RMSE, $R^2$ coefficients for both elastic net and multiple regression are presented to demonstrate elastic net's ability to mitigate model overfitting.

Variable Importance Plots (VIP) are used to determine coefficient significance due to elastic net regression's lack of theoretically derived $p$-values. Importance is calculated based on the absolute value of the coefficients [25]. Plots include both the individual variable importance values for each HHS region per time period, as well as box plots to view the average

**Table 2. HHS region numbers and their member states used in all analyses.**

| Number | States |
|---|---|
| 1 | CT, MA, ME, NH, RI, VT |
| 2 | NJ, NY |
| 3 | DE, MD, PA, VA, WV |
| 4 | AL, FL, GA, KY, MS, NC, SC, TN |
| 5 | IL, IN, MI, MN, OH, WI |
| 6 | AR, LA, NM, OK, TX |
| 7 | IA, KS, MO, NE |
| 8 | CO, MT, ND, SD, UT, WY |
| 9 | AZ, CA, NV |
| 10 | ID, OR, WA |

importance of each explanatory variable. From the full set of 15 explanatory variables, the five variables which exhibited the most noteworthy correlations with cases are isolated and interpreted. S1 and S2 Tables contain the full models with all 15 explanatory variables. Plots of the observed data and the associated predicted regression line for each region are also provided to assess individual model performance.

## Results

All coefficient estimates are reported in their original scale, allowing for direct interpretation. Overall, the RMSEs for elastic net models of the test data sets were lower than their multiple regression counterparts in all but two of the regions (HHS region 5 for pre-Delta, HHS region 6 for Delta); this reduction in RMSE indicates more robust model performance for the elastic net models. Individual model attributes will be discussed in their respective subsections.

The pre-Delta coefficient estimates and metrics are shown in Table 3. In regions 3, 6, and 10, elastic net regression removes several features from the regression model (see S1 Table). The small difference in training/testing $R^2$ for elastic net regression (Table 3) demonstrates that this method of regression avoids the problem of model overfitting. Fig 1 displays the VIP plot for the pre-Delta and Delta COVID-19 time periods. As shown in Fig 1a, we find that the percentage of individuals living in group quarters has the highest overall variable importance across all regions, with Democratic voting percentage and percentage of individuals in mobile homes being the other most important variables, respectively (Fig 1a). Of the selected variables, percentage non-institutionalized disabled individuals is least significant.

The Delta coefficient estimates and metrics are shown in Table 4. While there is some change in the features selected by the elastic net regression models in this time period, the $R^2$ comparisons for the testing and training data sets show that the method effectively avoids the problem of overfitting. Fig 1b displays the VIP plots for the Delta time period. The variable of lowest importance in the Delta time period is the percentage of individuals living in group quarters. Democratic voting percentage has the highest variable importance across all regions, followed by the percentage of households living in mobile homes and the percentage of

**Table 3. Coefficients and metrics for the 10 HHS regions for the pre-Delta COVID-19 time period, recorded from March 22, 2020 to June 15, 2021.**

| | | | | | Coefficients | | | | | |
|---|---|---|---|---|---|---|---|---|---|---|
| Region | 1 | 2 | 3 | 4 | 5 | 6 | 7 | 8 | 9 | 10 |
| (Intercept) | 65.14 | 87.20 | 85.06 | 109.20 | 106.13 | 107.57 | 106.38 | 113.58 | 88.21 | 71.52 |
| Disability | −2.95 | – | – | – | 3.21 | – | −0.84 | −4.60 | −6.11 | – |
| Minority | 3.71 | 4.59 | – | 5.07 | 8.18 | – | −1.13 | – | 24.01 | – |
| Mobile Housing | −12.5 | −7.95 | – | −6.16 | −3.67 | −2.85 | −6.61 | −7.19 | – | −1.03 |
| Group Quarters | −0.59 | −3.61 | 11.00 | 6.66 | 5.29 | 6.43 | 6.21 | 17.06 | 13.59 | 4.11 |
| Voting Percentage | −5.68 | −3.99 | −8.33 | −9.38 | −10.50 | −0.87 | −1.07 | −1.77 | −11.21 | −9.85 |
| | | | | | Metrics | | | | | |
| $\alpha$ | 0.10 | 0.19 | 0.50 | 0.91 | 0.36 | 0.54 | 0.73 | 0.81 | 0.27 | 0.42 |
| $\lambda$ | 4.31 | 2.61 | 0.46 | 0.20 | 0.09 | 2.10 | 0.37 | 0.99 | 0.81 | 3.58 |
| ENR Train $R^2$ | 0.78 | 0.78 | 0.44 | 0.20 | 0.41 | 0.28 | 0.35 | 0.40 | 0.71 | 0.66 |
| ENR Test $R^2$ | 0.78 | 0.78 | 0.42 | 0.15 | 0.37 | 0.24 | 0.35 | 0.37 | 0.64 | 0.64 |
| MR Train $R^2$ | 0.82 | 0.84 | 0.50 | 0.22 | 0.39 | 0.30 | 0.38 | 0.48 | 0.73 | 0.75 |
| MR Test $R^2$ | 0.76 | 0.63 | 0.34 | 0.11 | 0.32 | 0.20 | 0.30 | 0.25 | 0.46 | 0.60 |
| ENR Test RMSE | 15.48 | 11.52 | 12.88 | 20.00 | 15.68 | 23.95 | 21.05 | 28.11 | 26.79 | 19.27 |
| MR Test RMSE | 17.49 | 16.66 | 13.12 | 20.09 | 15.67 | 33.60 | 21.10 | 29.25 | 28.43 | 19.52 |

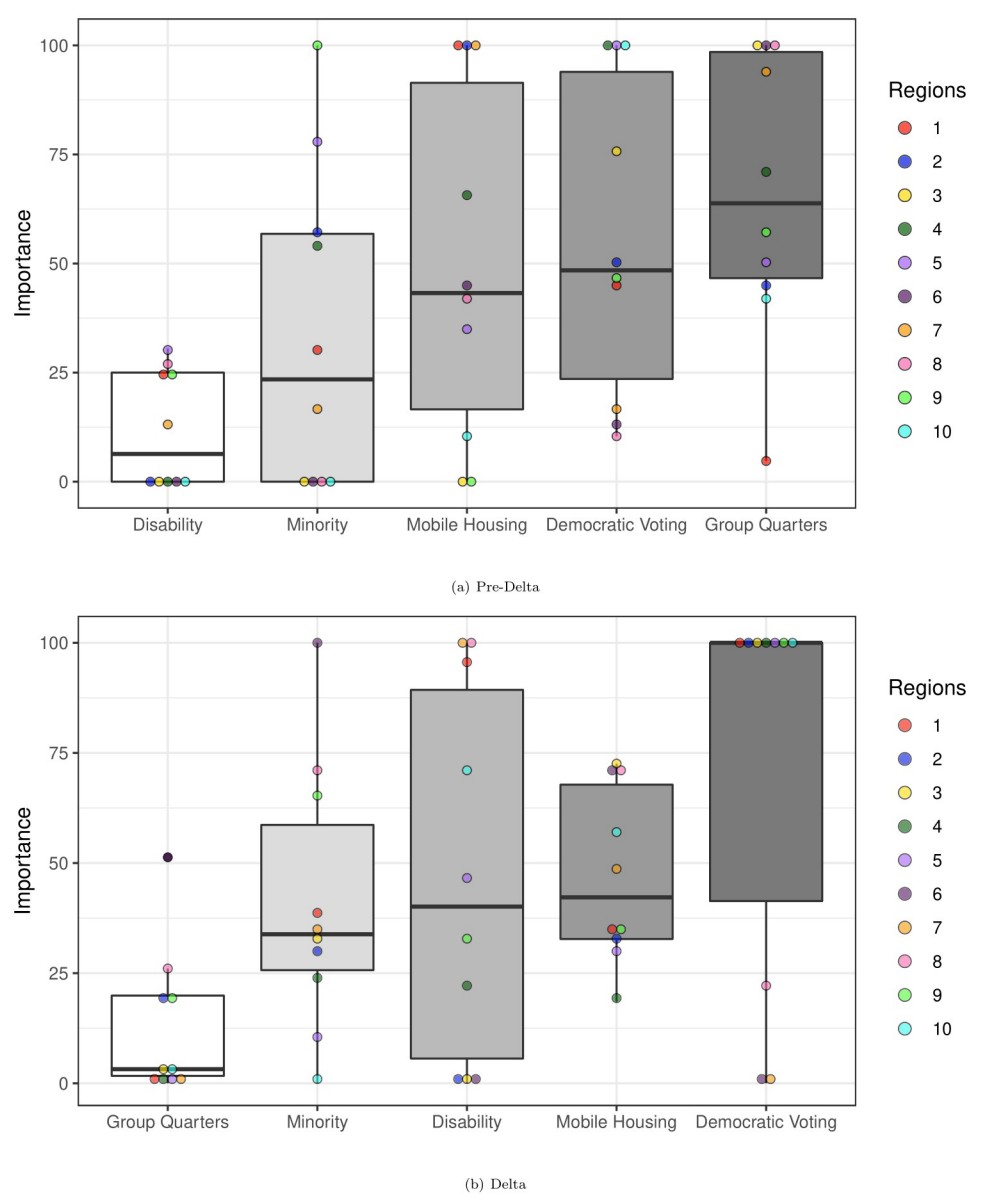

**Fig 1. Variable importance plots for both pre-Delta and Delta time periods, organized from lowest to highest overall importance.** A: Pre-Delta. B: Delta.

disabled individuals per county. Fig 1b shows that voting percentage has the highest individual variable importance in seven of the ten HHS regions for the Delta time period. Of the selected variables, group quarters has the largest shift in importance, going from most important to least important, whereas disability percentage increases dramatically in significance from pre-Delta to Delta.

Fig 2 displays the model fit for both pandemic time periods across each region, with each individual point having opacity and size based on population density (e.g., a county with low population density will have a small, transparent data point, and vice versa for a county with high population density). Regions 1, 2, 3, 9, and 10 (i.e., regions along the coast) present much better overall model fit when compared to the other five HHS regions (i.e., the inland regions),

**Table 4. Coefficients and metrics for the 10 HHS regions for the Delta COVID-19 time period, recorded from June 15, 2021 to November 1, 2021.**

| | Coefficients | | | | | | | | | |
|---|---|---|---|---|---|---|---|---|---|---|
| Region | 1 | 2 | 3 | 4 | 5 | 6 | 7 | 8 | 9 | 10 |
| (Intercept) | 23.50 | 24.60 | 49.69 | 61.08 | 40.32 | 49.07 | 36.03 | 39.92 | 40.47 | 46.76 |
| Disability | 1.31 | – | – | 2.20 | 3.77 | – | 3.76 | 6.64 | 2.57 | 4.16 |
| Minority | −0.53 | −0.69 | −2.30 | −2.34 | 0.85 | −1.91 | −1.34 | −4.76 | −4.99 | – |
| Mobile Housing | 0.49 | −0.72 | 5.21 | 1.81 | 2.35 | 1.37 | 1.83 | 4.74 | −2.61 | 3.37 |
| Group Quarters | – | −0.45 | −0.29 | 0.19 | 0.13 | −0.98 | – | −1.73 | −1.46 | 0.14 |
| Voting Percentage | −1.37 | −2.23 | −7.18 | −9.78 | −8.09 | – | – | −1.45 | −7.64 | −5.91 |
| | Metrics | | | | | | | | | |
| $\alpha$ | 0.30 | 0.84 | 0.71 | 0.66 | 0.67 | 0.90 | 0.95 | 0.12 | 0.14 | 0.20 |
| $\lambda$ | 1.94 | 0.19 | 1.76 | 0.14 | 0.07 | 0.78 | 0.60 | 0.98 | 0.98 | 1.18 |
| ENR Train $R^2$ | 0.47 | 0.62 | 0.72 | 0.46 | 0.55 | 0.16 | 0.24 | 0.28 | 0.73 | 0.60 |
| ENR Test $R^2$ | 0.46 | 0.55 | 0.72 | 0.41 | 0.51 | 0.16 | 0.27 | 0.25 | 0.69 | 0.59 |
| MR Train $R^2$ | 0.61 | 0.72 | 0.84 | 0.47 | 0.57 | 0.21 | 0.31 | 0.30 | 0.76 | 0.70 |
| MR Test $R^2$ | 0.15 | 0.29 | 0.57 | 0.37 | 0.45 | 0.07 | 0.22 | 0.18 | 0.62 | 0.46 |
| ENR Test RMSE | 7.04 | 4.49 | 12.33 | 13.09 | 9.15 | 15.21 | 9.96 | 15.05 | 9.30 | 14.09 |
| MR Test RMSE | 12.05 | 4.77 | 14.83 | 13.16 | 9.18 | 14.92 | 10.36 | 16.04 | 9.79 | 14.86 |

as shown through each model's $R^2$ coefficient. Overall, most models fit the data well, being able to explain a fair proportion of the variation within noisy case data.

## Discussion

Elastic net regression improves the prediction of COVID-19 cases when compared to multiple regression. All but two of the HHS regions across both the pre-Delta and Delta time periods have a lower testing set RMSE when compared to their multiple regression counterparts. Model fit and $R^2$ coefficients vary based on region, with inland regions having lower $R^2$ values. The regions with lower $R^2$ values also happen to be ones with more counties of lower population density. Regions 1, 2, 3, 9, and 10 have a combined average population density of about 467 people per square mile, whereas Regions 4, 5, 6, 7, and 8 have a combined population density of about 115 people per square mile. This implies that, using our methods, relationships between the social vulnerability measures and disease burden are more difficult to observe in regions with lower population density. This is important to consider when planning for future mitigation of health emergencies.

It was known early on in the pandemic that close-quarters such as cruise ships were epicenters of COVID-19 spread [26]. Similarly, increasing the county-wide percentage of people living in group quarters, such as nursing homes and prisons, has also been shown to increase risk [27, 28]. Our results show that this continued to be important through the pre-Delta time period of the pandemic. For the pre-Delta time period, the percentage of individuals in group quarters has the highest average variable importance across all regions. In all but regions 1 and 2, there is a strong positive association with cases. However, group quarters has the lowest average variable importance for the Delta time period. There are several possible explanations for this. Vaccination roll-out early on in 2021 was shown to mitigate disease spread [29]. In many locales, scarce vaccination doses were prioritized for these populations and vaccine acceptance among the elderly and institutionalized was relatively high [3]. Additionally, the lower association between the percentage of residents living in group quarters and COVID-19 cases during the Delta time period could be explained by a larger initial attack rate within

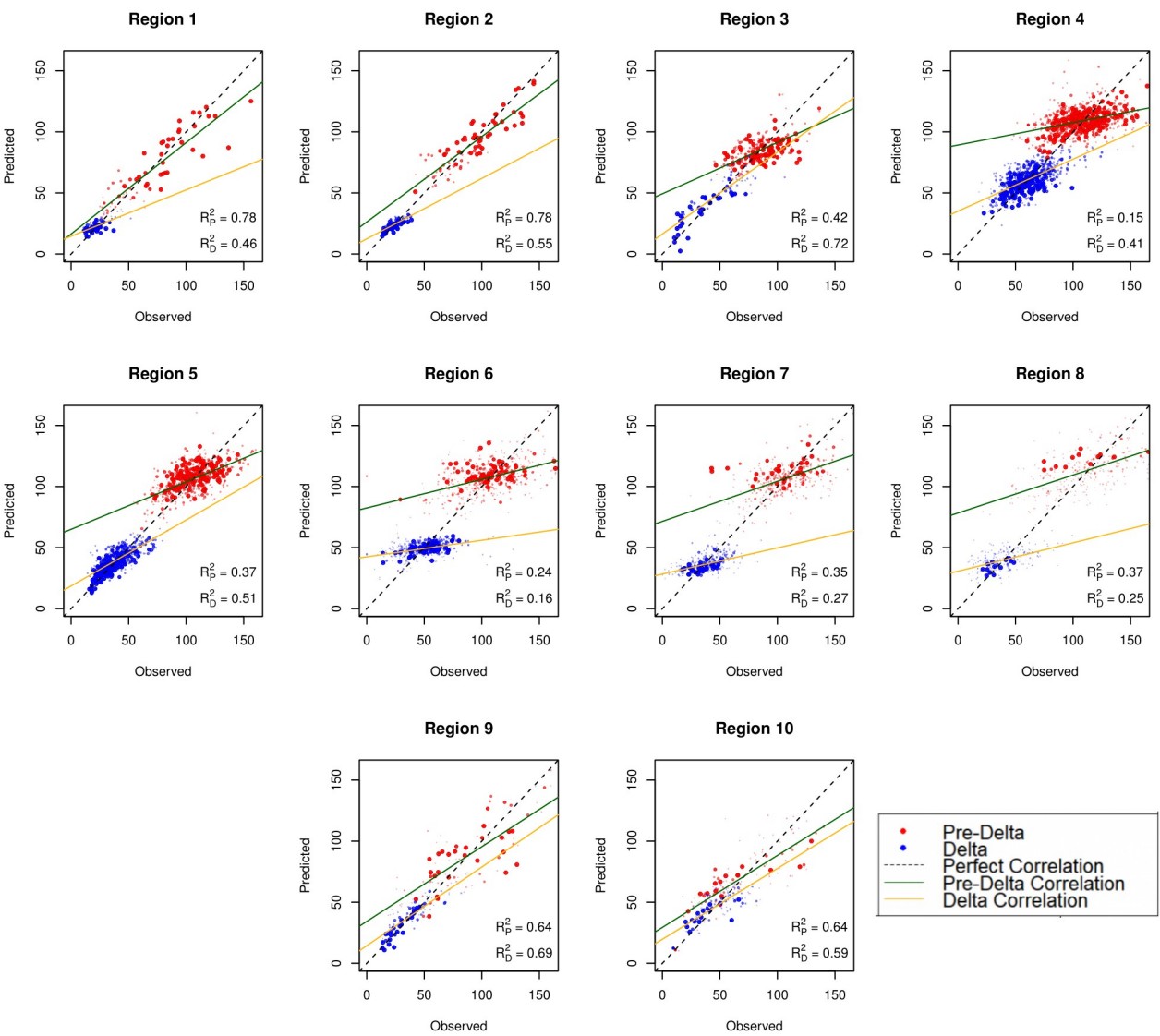

**Fig 2. Observed versus predicted COVID-19 case rate plot for both pandemic time periods, used to visualize model performance.** The "perfect" correlation line shows a perfect one-to-one relationship between the observed data and the fitted data, whereas each time period-specific line shows the observed correlation for each respective pandemic time period.

these counties. Indeed, many prisons were severely unequipped to mitigate COVID-19 transmission and thus suffered mass infections during the pre-Delta time period [30].

The percentage of non-institutionalized disabled persons per county was one of the least important variables in the pre-Delta time period and one of the most important variables in the Delta time period. The spectrum of what is considered disability is quite broad and encompasses many conditions (e.g., impairment of hearing, sight, mobility, or cognition) and thus the percentage of disabled people can be quite high (15.4%). Because the definition of disability by the American Community Survey (ACS) is broad, it makes our findings with respect to the percentage of disabled persons harder to interpret. The pre-Delta finding indicates the percentage of disabled people did not correlate with overall case counts. This is somewhat surprising; not only were many governments devoid of any resources for disabled individuals in their

pandemic responses, but several factors put disabled individuals at an increased risk for infection, such as dependence on caretakers creating often unavoidable exposure [31] and non-accessible hygiene and informational resources for individuals with visual, auditory, and cognitive disabilities [32]. However, the lack of relationship could either be due to the broad classification being used, or it could be that the fraction of individuals actually requiring specialized care is low enough that it did not affect the overall spread of COVID-19. We did observe the aforementioned trend of percentage of disabled persons predicting case counts in the Delta time period. It unclear why this would become a significant predictor later in the pandemic, however, it may be due to increased population mixing/reduced social distancing that occurred during this time period. If reduced social distancing is the cause, it might reflect that this population generally remained healthy early and thus acted as an influx of susceptible individuals later. It should be pointed out that looking at COVID-19 related deaths might have stronger relationship with the size of the disabled population, as the underlying comorbidities for disabled individuals present a higher likelihood of increased deaths rather than cases [33].

Percentage of minorities per-county was important across both pandemic time periods, with slightly higher importance in the Delta time period when transmissibility increased and mandated quarantining decreased. For the pre-Delta time period, all coefficients held positive values, which coincides with previous county-level research citing positive correlations with COVID-19 cases and minority proportions per county [34]. This is complementary to previous research citing a disproportionate impact of COVID-19 on minority individuals due to heightened levels of comorbidity, general inability to distance, and worse living conditions on average [8]. However, the trend reverses in the Delta time period, with many of the HHS regions having negative coefficient values. Like with group quarters, this could be due to a disproportionately high attack rate during the first major wave of the pandemic. Much like with disabled individuals, an analysis that takes into account COVID-19 deaths and hospitalizations may give more credence to the disproportionate toll of COVID-19 on counties with a higher proportion of minorities, given that many comorbidities minorities experience result in more severe infections regardless of SARS-CoV-2 variant [7].

Democratic voting percentage had highly significant negative correlations with cases across both pandemic time periods. This is supported by previous studies which cite a county-level partisan correlation (as determined by voting percentage) with respect to COVID-19 distancing [35], cases [36], and deaths [37]. Additionally, these findings complement previous research which cites a partisan divide with respect to trust in pandemic information resources and the effect it had on COVID-19 attack rates [38]. Given that vaccines were highly effective in reducing case incidence up through the Delta variant [39] and that Republican individuals reportedly had 90% lower odds of vaccination when compared to Democrats [40], disease behavior along partisan lines appears to have a significant effect on disease spread at the county level. Presidential voting is not the most foolproof proxy for the overall effect of political leaning; it fails to capture much of the extrinsic social effects caused by such factors as local legislature, personal risk evaluation, and media influence. However, it is still useful for potential policy inferences due to its availability and census-level information.

When considering previous literature, an important contribution of this approach is our submodeling structure through HHS regions. The purpose of HHS regional offices is to allow for better aid in local- and state-level healthcare facilities within their respective regions [23]. COVID-19 attack rate was known to vary across the United States due to differences in health infrastructure [24], which we observed through submodeling. An example of this can be visualized in Table 3, which shows a negative correlation between group quarters and COVID-19 case rate in Regions 1 and 2 and positive correlations elsewhere. Even without significance testing for elastic net, a change in the sign of a predictor's coefficient is significant enough to

assume spatial differences in pandemic response infrastructure. From a literature perspective, we know that differences in group quarters infrastructure and infection measures can affect the risk of secondary transmission [41]. Thus, our submodels are capable of capturing spatial variation in certain social risk factors that otherwise would not have been noticed. This both gives a retrospective into which areas of the US had better infrastructure and prophylactic response to COVID-19, as well as provides the framework for regional-level policy adjustments in future pandemics. Using categorical variables specific to region (e.g., state, HHS region, time zone, etc.) can only serve to tell us if spatial variation exists, but cannot tell us *how* it varies. Thus, not only did we use census data that observes county-level social burden for specific use in emergency situations, but we also give insight into spatial variation that can be evaluated by entities like the HHS to modify pandemic policy at a regional level [17, 23].

Additionally, our research expands the scope of social risk and its impact on COVID-19, both across a more representative time period and with more sophisticated methodology. The research done by [27, 28] share the most similarities to the analyses done here, with both utilizing SVI to determine disease burden. However, both attempts only represent COVID-19 cases through the first 3 to 4 months of the pandemic (the former only gathered data through July 29, 2020, and the latter through June 12, 2020). Our retrospective analyses give insight into COVID-19 social risk factors across not only a more representative period of the initial wave of COVID-19, but also of how those factors changed in the wake of new variants. Our methodology is also unique to social risk factor analysis due to its usage of regularization techniques to handle the implicit collinearity of SVI data. We know not only that social risk factors are often comorbid [7], but that all individual SVI metrics are potential variables of interest due to their inclusion in emergency and disaster response [17]. Both [27, 28] opted for negative binomial regression, which was appropriate for at the time given the zero-inflated distribution of early COVID-19 case data (e.g., many counties had 0 reported cases across the first few months of the COVID-19 pandemic) [1]. Importantly, our retrospective application considers the effect of collinearity in SVI variables, which was not part of the univariate regression models in [27]. Lastly, the use of elastic-net regression improves upon the practice of manually removing explanatory variables based on, e.g., Variance Inflation Factors or stepwise regression (as in [28]). Recent research [21] has shown that manual exclusion of variables introduces unwanted bias into final model result, and thus using regularization via elastic-net might preferable for model fitting because it can potentially retain collinear variables using shrinkage estimators [19].

## Limitations

Although our methods and data were carefully chosen, some limitations still exist. The analysis conducted on the pre-Delta time period had a much larger pool of data when compared to the Delta model, since the pre-Delta model contained 15 months worth of cumulative COVID-19 data, whereas the Delta model only contained five months. Further subdivision of COVID-19 time periods may be an appropriate approach for refining this research by separating the models out into the original strain, Alpha, Delta, Omicron, etc. Additionally, many social factors were not considered in this research. As alluded to when discussing political influences, vaccine hesitancy is a highly-important extrinsic social factor that has clear implications on local-level COVID-19 attack rates. Inclusion of vaccine hesitancy (or, at minimum, a proxy for this factor), would likely yield significant results. Another potential confounding variable is population density. Model fit discrepancies according to population density (Fig 2) may imply population density as an influential variable in spite of response variable standardization. However, healthcare infrastructure and subsequent differences in testing efforts likely

contribute to this effect, which is much more difficult to overcome via statistical methods alone [20, 24]. A more robust analysis using our modeling techniques would include more social factors of perceived importance as well as confounders such as population density. The addition of more explanatory factors (e.g., social factors, population density, healthcare measures) would most likely improve the predictive capabilities of our models; however, the goal of this work was to apply elastic-net regularization to collinear SVI measures at the administrative level of the HHS region to produce a robust, predictive model of disease burden.

We chose to use case rates (i.e., case count per 1000 people) as our response variable for the elastic net regression models. Standardizing the response variables in this method allows for use of standard multiple regression along with elastic net regression, as the case rates approximately follow a normal distribution (S1 Fig). Previous analyses by [27, 28], use similarly scaled COVID-19 cases/deaths (with [27] using cases per 100,000 and [28] using adjusted case-fatality rate). However, both of these analyses opt for using negative binomial regression via generalized linear models as the response variable is count data at its core. Thus, this analysis takes a different modeling approach when compared to other prior analyses. The net effect of choosing to use a normally distributed error versus some other model is a shift in parameter confidence intervals. Because we use variable importance to judge the reliability of our predictors, confidence intervals of parameters are not relevant.

Finally, we believe that these analyses may benefit from using deaths or hospitalizations as the response variable, rather than case counts. As seen with group quarters and minorities per county, the overall course of case spread does not necessarily display the broader picture of disease severity according to the pre-existing research [8, 26]. Cases can only predict so much; vaccine and natural immunity can suppress the perceived effect on individuals who are either at-risk of infection due to their environment, or are at-risk of severe illness due to autoimmune disorders or comorbidities. The application of this type of method to hospitalizations and deaths can help to enhance these results. In general, more consideration in using additional nuanced predictors along with repeating the analyses with hospitalizations and/or deaths would be valuable when thinking about future disease response.

## Conclusion

Understanding the relationship between COVID-19 case rates and existing measures of social vulnerability can help inform future pandemic planning. We found a mixture of intrinsic (disability and minority percentages in counties) and extrinsic (Democrat voting, mobile housing, and group quarters percentages) social factors. Policy decisions for emerging pandemics have two avenues: resource allocation (e.g., vaccines, masks, etc.) to areas that have higher attack rates due to intrinsic factors, and implementing measures (e.g., information campaigns) prior to disease outbreaks to better prepare vulnerable areas with higher extrinsic risk factors. In other words, our results suggest physical pandemic resources should be allocated toward counties with higher social vulnerability measures such as those with larger proportions of minorities or disabled individuals. Conversely, it has been observed that the political climate within the United States diminished the trustworthiness of many public information outlets like CDC [38]. Our findings, as well as those of others, suggest an appropriate prophylactic measure would be establishing trusted information sources in preparation for future outbreaks. Providing a unified, trusted, non-partisan outlet for pandemic information can ensure the public receives accurate messages regarding disease prevention. Future research may involve either expanding the feature set provided in the analysis to better understand the county-level correlations present in COVID-19 case data, or in looking more deeply into the significant variables found within this analysis to better understand the correlations.

## Supporting information

**S1 Fig. Case probability distributions and heat maps for both pre-Delta and Delta pandemic time periods (heat maps generated using usmap package in R version 4.1.3 [42]).** A: Pre-Delta Distribution. B: Pre-Delta Heat Map. C: Delta Distribution. D: Delta Heat Map.
(TIFF)

**S2 Fig. Display of varying optimal hyperparameters across two iterations of generating the elastic net model for the Pacific Northwest HHS region.** A: Iteration 1. B: Iteration 2.
(TIFF)

**S3 Fig. Complete pre-Delta variable importance plot, organized from lowest to highest overall importance.**
(TIFF)

**S4 Fig. Complete Delta variable importance plot, organized from lowest to highest overall importance.**
(TIFF)

**S1 Appendix. Elastic net regression overview [19, 43].**
(PDF)

**S1 Table. Full model table for pre-Delta COVID-19 time period.**
(PDF)

**S2 Table. Full model table for Delta COVID-19 time period.**
(PDF)

## Author Contributions

**Conceptualization:** Erich Seamon, Benjamin J. Ridenhour.

**Data curation:** Erich Seamon.

**Formal analysis:** Tristan A. Moxley.

**Funding acquisition:** Benjamin J. Ridenhour.

**Investigation:** Jennifer Johnson-Leung, Erich Seamon, Benjamin J. Ridenhour.

**Methodology:** Tristan A. Moxley, Erich Seamon.

**Project administration:** Benjamin J. Ridenhour.

**Resources:** Erich Seamon.

**Supervision:** Jennifer Johnson-Leung, Christopher Williams, Benjamin J. Ridenhour.

**Validation:** Jennifer Johnson-Leung, Christopher Williams, Benjamin J. Ridenhour.

**Visualization:** Tristan A. Moxley.

**Writing – original draft:** Tristan A. Moxley.

**Writing – review & editing:** Jennifer Johnson-Leung, Christopher Williams, Benjamin J. Ridenhour.

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
