## [Decision Letter · Decision Letter 0]

20 Sep 2023

PONE-D-23-03675Application of Elastic Net Regression for Modeling COVID-19 Sociodemographic Risk FactorsPLOS ONE

Dear Dr. Moxley,

Thank you for submitting your manuscript to PLOS ONE. After careful consideration, we feel that it has merit but does not fully meet PLOS ONE’s publication criteria as it currently stands. Therefore, we invite you to submit a revised version of the manuscript that addresses the points raised during the review process. Both reviewers have provided their major and minor concerns of the paper, for which the editor found reasonable and important. A major revision according to the reviewers' comments is needed. Among all meaningful comments from the two reviewers, the editor wants to highlight the needs of this paper in discussions of true contribution of the paper, which is critical as it stands. That is, how the methodology, data and/or findings of the paper contributes to the existing literature. Please submit your revised manuscript by Nov 04 2023 11:59PM. If you will need more time than this to complete your revisions, please reply to this message or contact the journal office at plosone@plos.org. Please include the following items when submitting your revised manuscript:A rebuttal letter that responds to each point raised by the academic editor and reviewer(s). You should upload this letter as a separate file labeled 'Response to Reviewers'.A marked-up copy of your manuscript that highlights changes made to the original version. You should upload this as a separate file labeled 'Revised Manuscript with Track Changes'.An unmarked version of your revised paper without tracked changes. You should upload this as a separate file labeled 'Manuscript'.

We look forward to receiving your revised manuscript.

Kind regards,

Chenfeng Xiong

Academic Editor

PLOS ONE

Journal Requirements:

2. We note that Figure S1 in your submission contain [map/satellite] images which may be copyrighted. All PLOS content is published under the Creative Commons Attribution License (CC BY 4.0), which means that the manuscript, images, and Supporting Information files will be freely available online, and any third party is permitted to access, download, copy, distribute, and use these materials in any way, even commercially, with proper attribution. For these reasons, we cannot publish previously copyrighted maps or satellite images created using proprietary data, such as Google software (Google Maps, Street View, and Earth). For more information, see our copyright guidelines: http://journals.plos.org/plosone/s/licenses-and-copyright.

 a. You may seek permission from the original copyright holder of Figure S1 to publish the content specifically under the CC BY 4.0 license. 

Additional Editor Comments (if provided):

A major revision according to the reviewers' comments is needed. Among all meaningful comments from the two reviewers, the editor wants to highlight the needs of this paper in discussions of true contribution of the paper, which is critical as it stands. That is, how the methodology, data and/or findings of the paper contributes to the existing literature.

Reviewers' comments:

Reviewer's Responses to Questions

**Comments to the Author**

1. Is the manuscript technically sound, and do the data support the conclusions?

Reviewer #1: Partly

Reviewer #2: Partly

2. Has the statistical analysis been performed appropriately and rigorously? 

Reviewer #1: Yes

Reviewer #2: Yes

3. Have the authors made all data underlying the findings in their manuscript fully available?

Reviewer #1: Yes

Reviewer #2: Yes

4. Is the manuscript presented in an intelligible fashion and written in standard English?

Reviewer #1: Yes

Reviewer #2: Yes

5. Review Comments to the Author

Reviewer #1: The paper applies elastic net regression to analyze social factors’ influences on COVID-19 cases in U.S. The models and results are carefully demonstrated. The implications and limitations are also thoroughly discussed. However, the contributions are not clear enough. Considering a good amount of previous studies on similar topics (relationship of COVID cases and socio-demographics), the paper should highlight the similarities and differences of this study. How it provides new information or better practice to the knowledge base? Since it seems not innovative in terms of methodology or data.

Minor comments:

Why choosing the HHS regions? Will other groupings of states/counties make more sense?

The model fit fluctuates a lot across regions. The population density was discussed as one factor. But no further evidence is provided. The transferability of the model deems questionable.

Reviewer #2: Thank you for the opportunity to review the manuscript entitled “Application of Elastic Net Regression for Modeling COVID-19 Sociodemographic Risk Factors”. The article is of interest for the readership of PLOS ONE as it focuses on factors of influence for case rate of COVID-19.

First, in my opinion, the authors failed to explain the reason for which constructing 10 models (one per region) is a preferred approach to 1 general model - for each time frame. While it is clearly explained why the authors split the analysis into the 2 time periods, the option of looking separately at each region is not argued in a proper manner.

Second, the variations in terms of health care infrastructure as a proxy for preventive measures could be also a relevant variable of influence. The authors should consider discussing it (at least in the limitation section).

I hope that my comments are useful for authors, as they further develop the study.

6. PLOS authors have the option to publish the peer review history of their article (what does this mean?). If published, this will include your full peer review and any attached files.

Reviewer #1: No

Reviewer #2: No

---

## [Author Response · Author response to Decision Letter 0]

7 Dec 2023

Dear editor(s),

We would like to thank the reviewers and editors for their helpful comments on our manuscript. In italic below, we give our responses to the editorial requests and the reviewers comments. In particular, we focus on framing our results in the current body of literature on COVID-19 and social risk to emphasize the contribution of this paper. We also give further justification of our use of HHS regional models by pointing out the significant spatial variation of the US with respect to both the SVI measures and COVID-19 cases. Additionally, we emphasize the role of HHS regions in determining healthcare policy and infrastructure. We now highlight comparisons with other work using SVI measures to study COVID-19 in the US, emphasizing that our approach considers inherent collinearity in the data structure of sociodemographic variables in addition to providing a retrospective account across two complete waves of the COVID-19 pandemic. Finally, we clarified limitations regarding discrepancies in model fit by HHS region, providing further context to these differences by citing data quality and testing efforts as potential confounding variables. We hope that our work is now suitable for publication and thank you again for your consideration of our work for publication in PLOS ONE.

Best,

Tristan Moxley

Editorial Requests:

1. We believe all of our materials match the PLOS ONE styles that are specified.

2. We are confused by the response that Figure S1 violates copyright. We generated this figure ourselves using publicly available COVID-19 data and the R package usmap, a subset of ggplot2. We have provided a citation for this image for clarification purposes, but wanted to express that this image was original to our research.

Reviewer #1: 

The paper applies elastic net regression to analyze social factors’ influences on COVID-19 cases in U.S. The models and results are carefully demonstrated. The implications and limitations are also thoroughly discussed. However, the contributions are not clear enough. Considering a good amount of previous studies on similar topics (relationship of COVID cases and socio-demographics), the paper should highlight the similarities and differences of this study. How it provides new information or better practice to the knowledge base? Since it seems not innovative in terms of methodology or data.

RESPONSE 1: We added a paragraph on Line 257 which serves to distinguish our work from other literature which adopted a similar framework for analysis. To summarize, we note that our research retrospectively considers a larger time frame when compared to other similar analyses. Our approach is also the only one to our knowledge that employs elastic net regression specifically for COVID-19 risk factor analysis. We conducted a Web of Science search on 26 October 2023. The search “elastic-net, COVID-19, SVI” returned zero hits. The search “elastic-net, COVID-19, socio-demographic" returned 3 hits, with one using closely related methods to study COVID-19 burden in Malaysia. The search “elastic-net, COVID-19" returned 61 hits, with none related to disease burden in the US. We explain the value of elastic-net regression as an analytical tool, especially when considering both the inherent collinearity of SVI data, as well as the presumed need to consider all SVI metrics as variables of interest due to their inclusion in a disaster response database. In short, we clarify that our analyses take into careful consideration the statistical assumptions that are intrinsic to the data while presenting a novel approach to mitigate them. 

Minor comments:

Why choosing the HHS regions? Will other groupings of states/counties make more sense?

RESPONSE 2: Our choice of HHS regions was done to capture a proxy of healthcare infrastructure, give retrospective insight into the regional differences in pandemic response, and provide a basis for regional-level pandemic response policy. Grouping by HHS region allows for easier application of our results, as HHS regional offices make jurisdictions regarding policy within their respective regions. We add clarification to this point in a paragraph on Line 236 and provided context to this statement in the Methods and Materials section on Line 94.

The model fit fluctuates a lot across regions. The population density was discussed as one factor. But no further evidence is provided. The transferability of the model deems questionable.

RESPONSE 3: Spatial variation in the importance of the SVI measures for county-level COVID-19 disease burden is one reason for using regional models in our study. We also note that testing efforts and subsequent case reporting/data quality differed widely among the regions and could have a significant effect on model fit. In the sentence beginning on Line 295, we provide additional citations from recent literature regarding the effects of these model performance on Line 297.

Reviewer #2: 

Thank you for the opportunity to review the manuscript entitled “Application of Elastic Net Regression for Modeling COVID-19 Sociodemographic Risk Factors”. The article is of interest for the readership of PLOS ONE as it focuses on factors of influence for case rate of COVID-19.

First, in my opinion, the authors failed to explain the reason for which constructing 10 models (one per region) is a preferred approach to 1 general model - for each time frame. While it is clearly explained why the authors split the analysis into the 2 time periods, the option of looking separately at each region is not argued in a proper manner.

RESPONSE 4: As mentioned in RESPONSE 2, above, we have provided more context into our usage of HHS regions in Lines 236 and 94. Line 251 gives our rationale for using 10 submodels as opposed to a single nationwide model specifically:

“Using categorical variables specific to region (e.g., state, HHS region, time zone, etc.) can only serve to tell us if spatial variation exists, but cannot tell us how it varies.”

Second, the variations in terms of health care infrastructure as a proxy for preventive measures could be also a relevant variable of influence. The authors should consider discussing it (at least in the limitation section).

RESPONSE 5: It is true that using more granular explanatory variables (e.g., characteristics of the HHS regions like number of hospital beds) would likely produce a better predictive model. However, the purpose of our research was to apply a cutting-edge regularization method (elastic net) to a novel application (predicting disease burden) using measures developed by DHHS/CDC for public health decisions during disaster responses. Thus, our goal isn’t necessarily to produce the very best predictive model of COVID-19 burden (which is almost an impossible task); rather, we wish to produce the most robust and predictive model using the collinear SVI measures. That being the case, breaking models down by HHS regions eliminates confounding due to broad-scale differences in healthcare infrastructure. We now clarify in the paragraph on Line 94 that the HHS submodels provide a proxy for healthcare infrastructure variance across the United States. Additionally, we now discuss this specific issue and our goals at the end of the first paragraph of the Limitations section.

---

## [Decision Letter · Decision Letter 1]

28 Dec 2023

Application of Elastic Net Regression for Modeling COVID-19 Sociodemographic Risk Factors

PONE-D-23-03675R1

Dear Dr. Moxley,

We’re pleased to inform you that your manuscript has been judged scientifically suitable for publication and will be formally accepted for publication once it meets all outstanding technical requirements.

Kind regards,

Chenfeng Xiong

Academic Editor

PLOS ONE

Additional Editor Comments (optional):

Reviewers' comments:

Reviewer's Responses to Questions

**Comments to the Author**

1. If the authors have adequately addressed your comments raised in a previous round of review and you feel that this manuscript is now acceptable for publication, you may indicate that here to bypass the “Comments to the Author” section, enter your conflict of interest statement in the “Confidential to Editor” section, and submit your "Accept" recommendation.

Reviewer #1: All comments have been addressed

Reviewer #2: All comments have been addressed

2. Is the manuscript technically sound, and do the data support the conclusions?

Reviewer #1: Yes

Reviewer #2: Yes

3. Has the statistical analysis been performed appropriately and rigorously? 

Reviewer #1: Yes

Reviewer #2: Yes

4. Have the authors made all data underlying the findings in their manuscript fully available?

Reviewer #1: Yes

Reviewer #2: Yes

5. Is the manuscript presented in an intelligible fashion and written in standard English?

Reviewer #1: Yes

Reviewer #2: Yes

6. Review Comments to the Author

Reviewer #1: All the comments were addressed. I would suggest highlight the novelty of the work not only in the discussion section but also in the introduction for publication.

Reviewer #2: Taking into account the revisions and the answer provided by the authors to my comments, I believe that the manuscript has been improved and now warrants publication.

7. PLOS authors have the option to publish the peer review history of their article (what does this mean?). If published, this will include your full peer review and any attached files.

Reviewer #1: No

Reviewer #2: No

---

## [Editor Report · Acceptance letter]

17 Jan 2024

PONE-D-23-03675R1 

PLOS ONE

Dear Dr. Moxley, 

I'm pleased to inform you that your manuscript has been deemed suitable for publication in PLOS ONE. Congratulations! Your manuscript is now being handed over to our production team.

Kind regards, 

on behalf of

Dr. Chenfeng Xiong 

Academic Editor

PLOS ONE